# Proteomic Analysis of Maternal Urine for the Early Detection of Preeclampsia and Fetal Growth Restriction

**DOI:** 10.3390/jcm10204679

**Published:** 2021-10-13

**Authors:** Emmanuel Bujold, Alexandre Fillion, Florence Roux-Dalvai, Marie Pier Scott-Boyer, Yves Giguère, Jean-Claude Forest, Clarisse Gotti, Geneviève Laforest, Paul Guerby, Arnaud Droit

**Affiliations:** 1CHU de Québec—Université Laval Research Center, Université Laval, Quebec City, QC G1V 4G2, Canada; alexandre.fillion.3@ulaval.ca (A.F.); florence.roux-dalvai@crchudequebec.ulaval.ca (F.R.-D.); marie-pier.scott-boyer.1@ulaval.ca (M.P.S.-B.); Yves.Giguere@crchudequebec.ulaval.ca (Y.G.); jean-claude.forest.med@ssss.gouv.qc.ca (J.-C.F.); clarisse.gotti@crchudequebec.ulaval.ca (C.G.); genevieve.laforest@crchudequebec.ulaval.ca (G.L.); paul.guerby@gmail.com (P.G.); arnaud.droit@crchudequebec.ulaval.ca (A.D.); 2Department of Obstetrics and Gynecology, Faculty of Medicine, CHU de Québec—Université Laval, Quebec City, QC G1V 4G2, Canada; 3Department of Molecular Biology, Medical Biochemistry and Pathology, Faculty of Medicine, CHU de Québec—Université Laval, Quebec City, QC G1V 4G2, Canada; 4Department of Obstetrics and Gynecology, Paule de Viguier Hospital, CHU de Toulouse—Université de Toulouse, 31059 Toulouse, France

**Keywords:** pregnancy, preeclampsia, proteomics, urine, biomarkers

## Abstract

Background: To explore the use of maternal urine proteome for the identification of preeclampsia biomarkers. Methods: Maternal urine samples from women with and without preeclampsia were used for protein discovery followed by a validation study. The targeted proteins of interest were then measured in urine samples collected at 20–24 and 30–34 weeks among nine women who developed preeclampsia, one woman with fetal growth restriction, and 20 women with uncomplicated pregnancies from a longitudinal study. Protein identification and quantification was obtained using liquid chromatography–tandem mass spectrometry (LC–MS/MS). Results: Among the 1108 urine proteins quantified in the discovery study, 21 were upregulated in preeclampsia and selected for validation. Nineteen (90%) proteins were confirmed as upregulated in preeclampsia cases. Among them, two proteins, ceruloplasmin and serpin A7, were upregulated at 20–24 weeks and 30–34 weeks of gestation (*p* < 0.05) in cases of preeclampsia, and could have served to identify 60% of women who subsequently developed preeclampsia and/or fetal growth restriction at 20–24 weeks of gestation, and 78% at 30–34 weeks, for a false-positive rate of 10%. Conclusions: Proteomic profiling of maternal urine can differentiate women with and without preeclampsia. Several proteins including ceruloplasmin and serpin A7 are upregulated in maternal urine before the diagnosis of preeclampsia and potentially fetal growth restriction.

## 1. Introduction

Preeclampsia (PE) affects about 2% to 5% of pregnant women in developed countries and up to 10% of pregnant women in developing countries [1,2,3]. PE is associated with short-term and long-term adverse outcomes in mothers and infants [4,5,6,7,8]. About 10% to 15% of maternal deaths and 25% of neonatal deaths are attributable to PE and its complications [8,9]. In the most severe cases, PE occurs before term and is typically associated with deep placentation disorders that can also lead to fetal growth restriction [10,11,12].

There is a growing body of evidence that the preterm PE, severe PE, and fetal growth restriction can be predicted in early pregnancy using a combination of biophysical, biochemical and ultrasound markers [13,14,15]. However, such screening tools require equipment and expertise that are not readily available throughout the world. There is evidence that urine biomarkers could also be used for the early diagnosis of PE [16,17].

Mass-spectrometry-based proteomics analysis has gained popularity in the past decades for its ability to cover a large proportion of cellular or biological fluid proteomes [18]. Indeed, bottom-up proteomics have allowed for the identification and quantification of hundreds of proteins. This strategy relies on the identification and quantification of peptides resulting from the trypsin digestion of protein extracts by liquid chromatography coupled with tandem mass spectrometry (LC–MS/MS). The acquired spectra are then searched against publicly available protein databases. Protein candidates can also be accurately quantified by targeted proteomics in which up to one hundred proteins can be specifically monitored by LC–MS/MS in a single analysis [19]. The urinary proteome, which can be collected non-invasively and which contains more than 1800 protein species, is an ideal source of biomarkers for both renal/urological tract and systemic diseases [20,21].

For preeclampsia, a recent review of proteomic studies reported that at least 132 urine proteins could be used for early diagnosis [22]. More importantly, Buhimschi et al. observed that a specific urine proteomic profile combining serpin A1 and albumin could have: (1) a high accuracy in the identification of women with severe PE that required immediate delivery; and (2) could identify women at high risk to develop severe PE up to 10 to 25 weeks before its diagnosis [16]. However, these observations were limited to a small number of participants; the experiments were performed using equipment developed many years ago, and should therefore be validated in larger studies.

In the current study, we aim to assess the urine proteomic profiles in PE and to identify biomarkers that could help in the early prediction of PE.

## 2. Materials and Methods

We performed a secondary analysis of the PEARL case-cohort study (PreEclampsia And growth Restriction: a Longitudinal study) that included both nulliparous women with PE (cases) and a cohort of nulliparous women at low risk of PE, recruited in early pregnancy and followed until delivery (controls) [23,24]. PE was defined according to the Society of Obstetricians and Gynecologists of Canada guidelines as de novo hypertension after 20 weeks of pregnancy (i.e., Systolic BP ≥ 140 mmHg and/or diastolic BP ≥ 90 mmHg) with proteinuria (≥300 mg per 24 h or ≥2 + protein on urine dipstick) or an adverse condition (thrombocytopenia, renal failure, liver injury, headache, seizures, vision problem or pulmonary edema) [25]. Fetal growth restriction was defined as a birth weight below the 10th percentile for gestational age based on a Canadian reference chart [26]. Urine samples were collected on the day of the diagnosis of PE for all cases and at several points during pregnancy for all controls. Urine samples were aliquoted and stored at −80 °C until analysis. All participants signed an informed consent form, and the study was approved by the ethics committee of the CHU de Québec—Université Laval (B14–07-2037).

For the purpose of the current study, we divided our population as follows: for the discovery study, 12 participants with PE (including 6 with fetal growth restriction) were matched to 12 controls without PE based on gestational age at urine collection, maternal age and maternal body mass index. For the validation study, another subset of 12 participants with PE (including 2 with fetal growth restriction) were matched to 12 controls also based on gestational age, maternal age and maternal body mass index. We used all samples available (*n* = 24) and divided by two for the discovery and validation study. For the longitudinal study, we used an independent cohort of women who were seen at each trimester of pregnancy: we selected 9 women who developed PE; 1 woman who developed fetal growth restriction without PE; and 20 women with uncomplicated pregnancies who provided urine samples at 20–24 and 30–34 weeks.

### 2.1. Mass Spectrometry Analyses

Each 500 µL sample of urine was concentrated on an Amicon Ultra-0.5 Centrifugal Filter device (Millipore, Burlington, MA, USA) by 10 min centrifugation 14,000× *g*, followed by a wash with 500 µL ammonium bicarbonate 50 mM and centrifugation in the same conditions. The protein concentration of the concentrated urine samples (volumes between 55 and 85 µL) was measured using Bradford assay. A total of 10 µg from each sample was then used for the subsequent steps. Briefly, the sample volume was adjusted to 50 µL with ammonium bicarbonate at 50 mM, and sodium deoxycholate was added to a final concentration of 1%. Protein denaturation was performed by heating at 95 °C for 5 min. The reduction and alkylation of cystein disulfide bridges was performed by the addition of 1,4 dithiothreitol (DTT) (final concentration 0.2 mM) and incubation at 37 °C for 30 min followed by the addition of iodoacetamide (final concentration 0.8 mM) and incubation at 37 °C for 30 min in the dark. An amount of 200 ng of tryspin enzyme was then added and samples were incubated at 37 °C overnight for proteolysis. Enzymatic digestion was stopped by acidification with 50% formic acid and the resulting peptides were purified on StageTip according to Rappsilber et al., using C18 Empore reverse phase (CDS) [27]. The samples were vacuum-dried and stored at −20 °C prior to mass spectrometry analysis. Each sample was resuspended at 0.2 µg/µL with 2% acetonitrile, 0.05% TFA. For validation and longitudinal studies only, iRT internal standard peptides (Biognosys, Schlieren, Switzerland) were added in each sample at 1X final concentration. An amount of 1µg from each sample was analyzed by liquid chromatography coupled with tandem mass spectrometry (LC–MS/MS) using a U3000 RSLCnano chromatographic system (Thermo Fisher Scientific, Waltham, MA, USA) and an Orbitrap Fusion mass spectrometer (Thermo Fisher Scientific). The chromatographic separation was performed on an Acclaim PepMap 100 C18 column (75 µm internal diameter, 3 µm particules and 500 mm length) using a 5–45% solvent B 90 min gradient (A: 5% acetonitrile, 0.1% formic acid; B: 80% acetonitrile, 0.1% formic acid).The mass spectrometer was operated in Data Dependent Acquisition mode for the discovery study, and Parallel Reaction Monitoring mode for the validation and longitudinal studies, using two peptide precursor masses for each protein selected from the discovery study. 

### 2.2. Bioinformatics and Statistical Treatment

For the discovery study, MaxQuant software was used to obtain protein identification by searching a Uniprot human database (Human Reference Proteome UP000005640), and quantification was obtained by the Label-Free Quantification (LFQ) method [28]. LFQ intensity values of MaxQuant were used to calculate a protein fold change (FC) between the two groups of patients, and the values were then centered by the calculation of a z-score (z = (FC − FC average)/FC standard deviation). Statistical analyses were then performed using R software [29]. A Limma statistical test was applied to each protein between the two groups and the corresponding *p*-values were adjusted for multiple testing using the Benjamini–Hochberg method and thus obtained *q*-values [30]. A protein was considered as significantly regulated between the two groups if it met the following criteria: |*z*| > 1.96 and *q* < 0.05. For the validation and longitudinal studies, the data analysis was performed using the Skyline software [31]. The quantification value of each targeted peptide was obtained by integration of the corresponding chromatographic peak reconstructed from the best parent/fragment transition. Two normalizations of the quantification values were then applied. The first used the median of iRT internal standard peptides intensity to correct for LC–MS/MS variability, the second used the signal of peptides from the protein biotinidase (BTD) (Uniprot accession number P43251), which displayed no variation between the two groups in our discovery analysis, to correct for Bradford protein assay. For each sample, the intensities of the two peptides of each protein were summed to obtain a protein intensity value. A modified statistical Student’s test (Welch test) was performed between the two groups of patients. Statistical analyses were conducted using R software and IBM SPSS Statistics version 26.0. Interaction network analysis was performed using the STRING database [32].

## 3. Results

### 3.1. Discovery Study

The median gestational age of the 12 participants at diagnosis of PE (31.9; IQ: 28.5–34.3 weeks) was comparable to the median gestational age of the 12 participants used as controls (32.1; IQ: 29.0–35.3 weeks). Table 1 reports participant’s characteristics for the discovery, the validation and the longitudinal study.

Our proteomic analyses allowed us to quantify 1108 proteins from urine samples which were used in a Principal Component Analysis (PCA) to explore the variability of the urinary proteomic profiles of participants (Figure 1). The PCA showed two distinct groups for control and PE revealing distinct proteomic profiles.

A heatmap of the intensity of the proteins quantified in this study was also generated and associated to a hierarchical clustering, which allowed us to group the most similar profiles among the urine proteomes (Figure 2).

Among all the quantified proteins, 62 (53 upregulated, 9 downregulated) were found statistically regulated between PE and control after filtering on z-score (obtained after centering of the protein fold change) and on the *q*-value associated to a Limma statistical test corrected for multiple testing (Figure 3). Twenty-one upregulated proteins were selected for further validation (A1BG; ALB; AFM; TTR; AZGP1; C3; CA1; CP; GC; HBA1; HBB; ITIH2; ORM1; ORM2; SERPINA1; SERPINA3; SERPINA6; SERPINA7; SERPINC1; SHBG; TF; see Table 2 for protein descriptions) based on their fold change, their total signal intensity and/or the sequence of their corresponding peptides (peptides carrying oxidized methionine and those containing missed trypsin cleavages were excluded). Using a STRING analysis, we found that most of them have known interactions, either direct protein interaction, gene co-expression, or are cited together in the literature (Appendix A).

### 3.2. Validation Study

The median gestational age at urine collection of our PE cases and term uncomplicated delivery controls was 31.5 (IQ: 29.0–34.3) weeks and 32.5 weeks (IQ: 29.6–33.9) weeks, respectively (Table 1). The 21 proteins previously selected were monitored by targeted proteomics to obtain an accurate protein quantification. Table 2 and Figure 4 report the average and distribution of intensity values for each group, PE or control, of each of the selected 21 proteins monitored by targeted proteomics. We observed that 19 (90%) of the 21 proteins were significantly upregulated in the urine of PE patients when compared to controls with a *p*-value < 0.05. Among them, 13 have a *p*-value < 0.001.

### 3.3. Longitudinal Study

In the longitudinal study, we observed that 6 of the 21 proteins monitored by targeted proteomics were significantly upregulated at 30–34 weeks in women who subsequently developed PE in comparison to the control group (serpin A7; ceruloplasmin; afamin; inter-alpha-trypsin inhibitor heavy chain; transferrin; alpha-1B-glycoprotein). Two, serpin A7 and ceruloplasmin, were also found significantly upregulated at 20–24 weeks in women who subsequently developed PE in comparison to the control group (Table 3 and Figure 5).

Figure 6A,B reports the receiver operating characteristic (ROC) curves for the prediction of PE or fetal growth restriction using ceruloplasmin and serpin A7 protein concentrations in urine at 20–24 weeks and 30–34 weeks of gestation. At a false positive rate of 10%, ceruloplasmin could have predicted between 60% and 78% of the PE and/or fetal growth restriction cases at 20–24 weeks and 30–34 weeks of gestation, respectively. Interestingly, the case of fetal growth restriction without preeclampsia would have been detected with ceruloplasmin but not with serpin A7.

## 4. Discussion

We observed that women with preeclampsia have a different urine proteomic profile than women without preeclampsia and such differences can be present up to 12 weeks before the first signs and symptoms of preeclampsia. This observation, consistent with previous studies, suggests that women at high-risk of preeclampsia could be detected from urine biomarkers. Since fetal growth restriction shares a common mechanism of disease, it is possible that it could be detected in early pregnancy as well.

More specifically, the PCA performed on our dataset discriminates the two groups of patients (PE vs. controls). Moreover, the hierarchical clustering associated with the heatmap of protein intensities generates two main clusters containing either control or PE samples. To reveal the proteins dysregulated between PE and controls, we centered the control/PE ratios by calculating a z-score. Indeed, the proteinuria associated to PE strongly affects the protein content of the analyzed samples. Therefore, lower mass spectrometry signals obtained in PE cases, as can be observed on the heatmap of protein expressions, might result in a misinterpretation of the dysregulated proteins between the two groups. By applying a z-score correction and using a statistical test corrected for multiple testing (*q*-values), we could confidently identify 62 proteins with statistical differences in intensity between control and PE groups. Using targeted proteomics, known for their high accuracy in protein quantification, and another subset of patients from the same cohort, we confirmed our findings from the discovery study. The monitoring of these 21 proteins in our longitudinal study revealed that 6 of these proteins were significantly upregulated at 30–34 weeks of gestation and 2 of them (ceruloplasmin and serpin A7) were significantly upregulated at 20–24 weeks of gestation, up to 12 weeks before the clinical onset of PE. Ceruloplasmin is a copper-binding protein involved in iron transport across cellular membranes, and has antioxidant ferroxidase properties. It could be upregulated in PE as a response to placental ischemia [33,34]. As for serpin A7, also known as thyroxine-binding globulin (TBG), the major thyroid hormone transport protein in serum, studies have observed lower serum TBG in women with PE [35,36]. However, little is known about how and why serpin A7 is upregulated in the urine of PE cases.

Overall, our observations suggest that the urine proteomic profile could be used to predict preeclampsia several weeks before the first signs and symptoms manifest themselves. More specifically, we observed that at least two proteins, namely, ceruloplasmin and serpin A7, are increased in maternal urine at 20–24 and 30–34 weeks of gestation in most women who subsequently develop PE.

Our study is in agreement with the study of Buhimschi et al., who observed that serpin A1, another member of the serpin family, was significantly increased in PE cases [16]. Their study also reported misfolded proteins in the urine of women with preeclampsia bound to Congo Red dye (urine congophilia or affinity for the amyloidophilic dye Congo Red) [37]. Ultimately, their study reported promising results from a prototype point-of-care test for the detection of urine congophilia [38]. A review by Navajas et al. found nine publications from 2008 to 2020 that observed 132 proteins that were differently expressed in urine in PE cases compared to controls [22]. Nineteen of these showed high potential for PE prediction as they were consistently higher or lower in PE, including ceruloplasmin, serpin A1, serpin A5, C3, ALB, TF and HBB. Starodubtseva et al. reported that the estimation of serpin A1 peptides in urine was also related to the severity of PE [39]. Placental growth factor (PlGF), a proangiogenic protein commonly used for the prediction of PE and other placental-mediated outcomes of pregnancy is also predictive of PE and PE-related adverse outcomes when measured in urine [17,23,40]. Altogether, these studies, including ours, provide hope that rapid identification of PE and potentially early prediction of PE is possible using urine studies.

The small number of cases available at each step is a limit of our study. However, we used standardized methods for proteomic analysis and, using high-resolution LC–MS/MS, we obtained the largest urinary proteome coverage ever published on preeclampsia samples (1108 proteins quantified). One of the major strengths of the current study is the validation of our findings in an independent subset of patients and subsequent confirmation in a prospective study, which includes the collection of urine samples up to three months before the first signs and symptoms of PE.

## 5. Conclusions

PE is a major cause of perinatal morbidity and mortality around the world, primarily in developing countries. PE is commonly associated with fetal growth restriction, as they share a common mechanism of disease. Our study and current literature strongly suggest that PE, and potentially fetal growth restriction, are syndromes that are highly detectable in their early phases using the proteomic analysis of maternal urine. More efforts should be devoted to the development of rapid point-of-care tests on maternal urine that could help in the prevention of PE-related adverse outcomes of pregnancy.

## Figures and Tables

**Figure 1 jcm-10-04679-f001:**
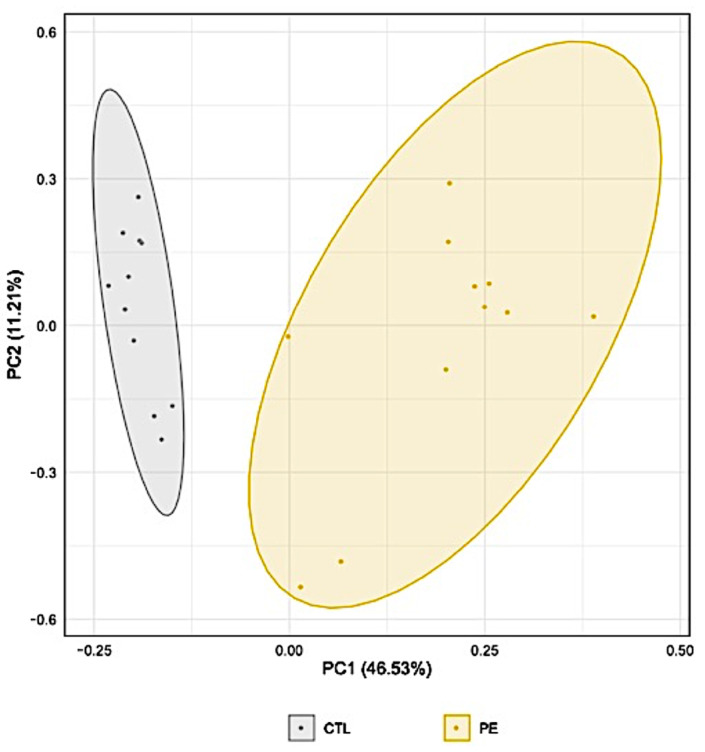
Principal Component Analysis (PCA) of the urinary proteomic profiles in participants with and without preeclampsia. Each point represents a participant. The axes correspond to artificial axes rebuilt by the PCA to display the maximum variability between the samples. CTL: Control; PE: Preeclampsia.

**Figure 2 jcm-10-04679-f002:**
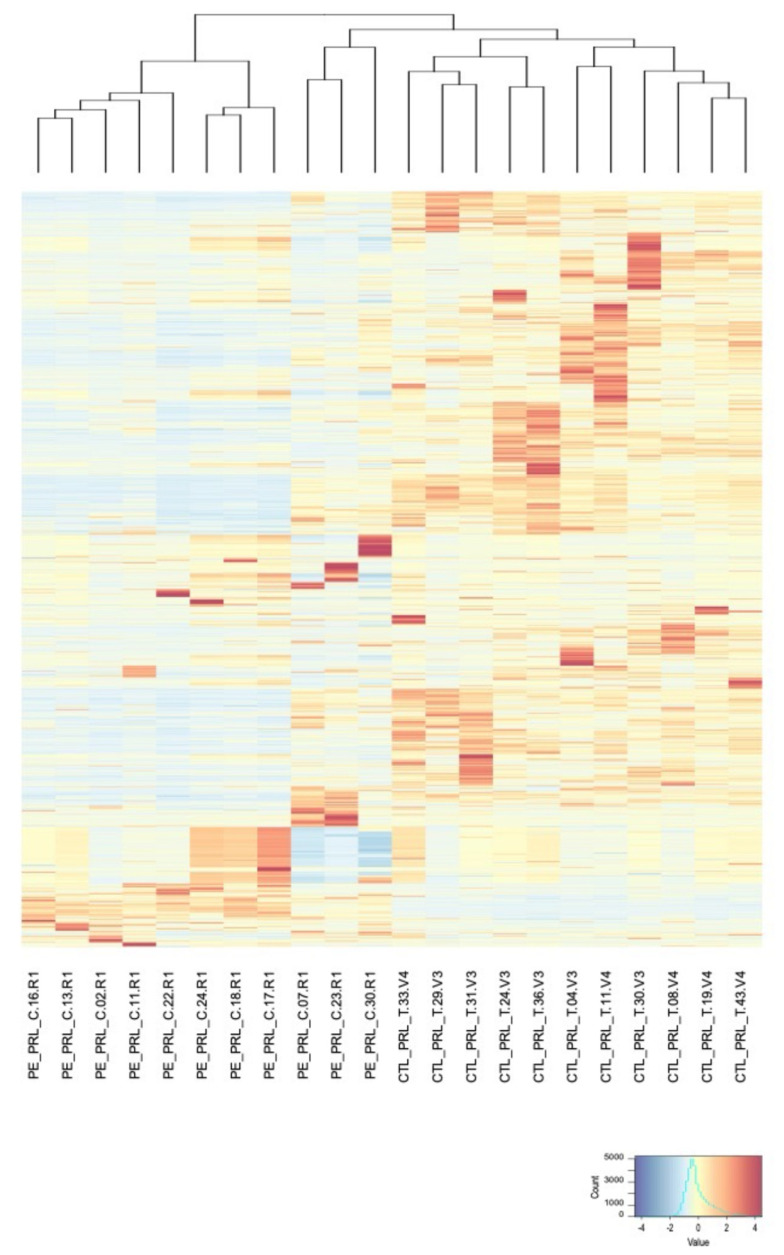
Heatmap of the expression of the 1108 proteins extracted from the proteomic analysis. Colors correspond to the centered intensity values of each protein in each urine sample from blue (smallest intensity) to red (highest intensity). The proteins (in lines) and the samples (in columns) are presented as clustered according to a complete linkage clustering method; the corresponding dendro-gram is shown only for samples.

**Figure 3 jcm-10-04679-f003:**
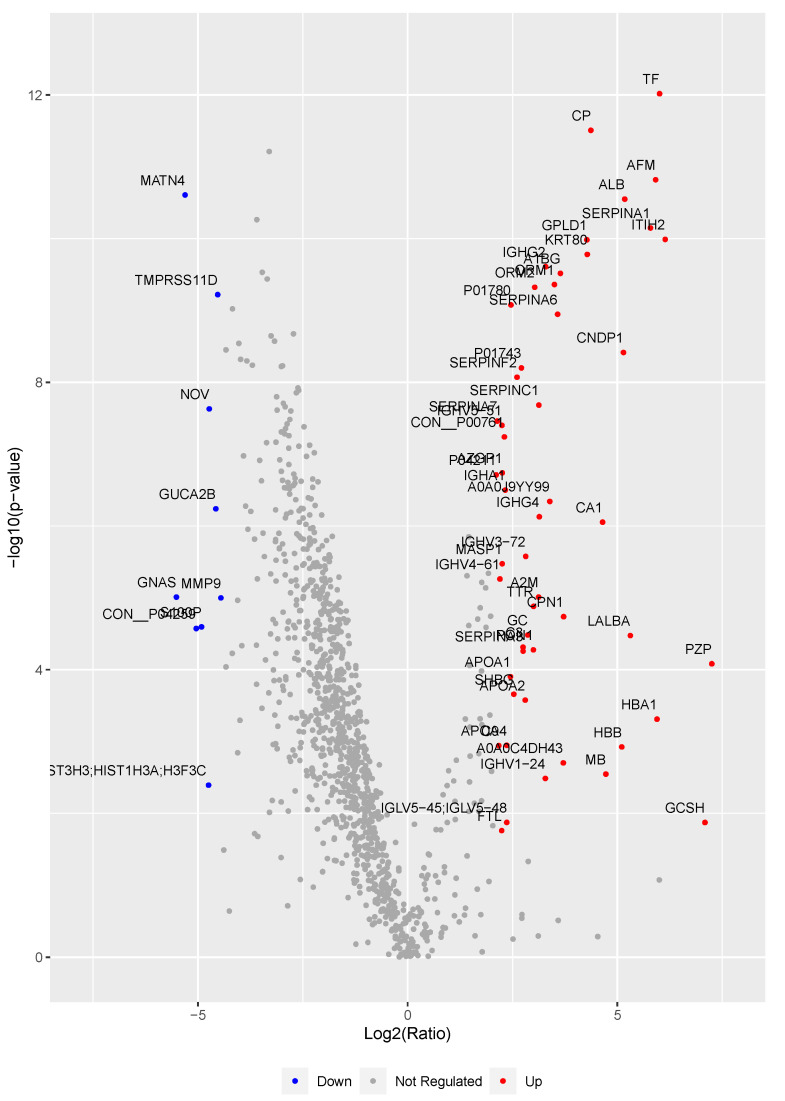
Volcano plot of the 1108 proteins quantified in the proteomic analysis. The x-axis corresponds to the fold change (Log2 of PE/control protein intensity ratio), the y-axis corresponds to the statistical value (−Log10 (*p*-value)). Significantly regulated proteins are displayed in red (upregulated) or in blue (downregulated); non-significantly regulated proteins are displayed in grey.

**Figure 4 jcm-10-04679-f004:**
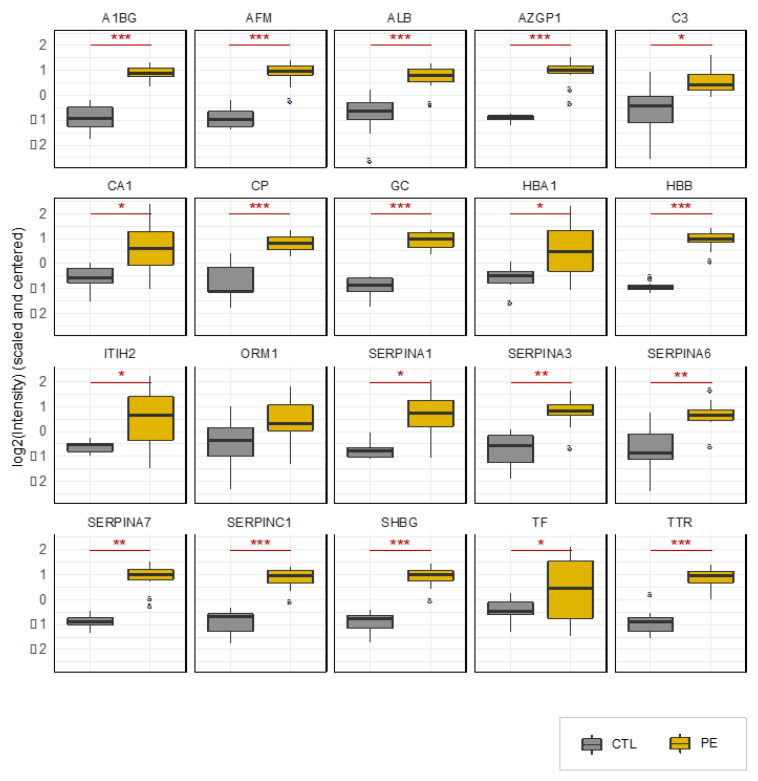
Validation study: Distribution of intensities measured by targeted proteomics of 21 urinary proteins in women with and without preeclampsia. The boxplots represent the distribution of intensities in each group (grey: control, yellow: PE) as well as the median (line), the interquartile (box) and the maximum and minimum values within 1.5 times the interquartile range (whisker) of 21 proteins monitored by targeted proteomics in urine samples of women with preeclampsia or women with uncomplicated pregnancies. Statistical significance is displayed above each graph: * *p* < 0.05; ** *p* < 0.01; *** *p* < 0.001.

**Figure 5 jcm-10-04679-f005:**
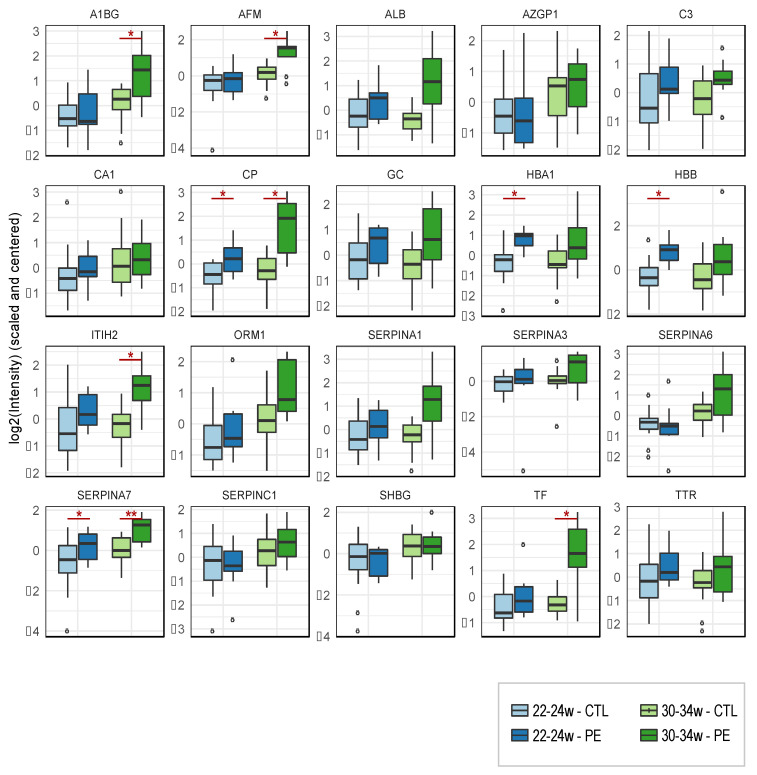
Longitudinal study: Distribution of intensities measured by targeted proteomics of 21 urinary proteins in women with and without preeclampsia at 20–24 or 30–34 weeks of gestation. The boxplots represent the distribution of intensities in each group as well as the median (line), the interquartile (box) and the maximum and minimum values within 1.5 times the interquartile range (whisker) of 21 proteins monitored by targeted proteomics in urine samples collected at 20–22 weeks (blue) and 30–32 weeks (green) of gestation in women with preeclampsia (*n* = 9) (dark color) or women with uncomplicated pregnancies (*n* = 20) (light color). Statistical significance is displayed above each graph: * *p* < 0.05; ** *p* < 0.01.

**Figure 6 jcm-10-04679-f006:**
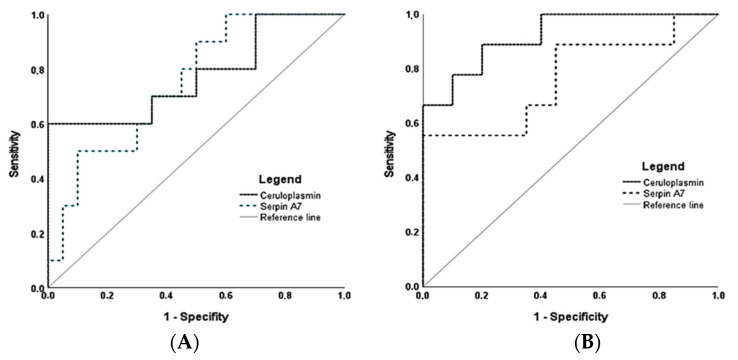
Receiver operating characteristic (ROC) curves for the prediction of preeclampsia and/or fetal growth restriction based on ceruloplasmin and serpin A7 proteins measured by targeted proteomics in urine of women at 20–24 and 30–34 weeks of gestation. The ROC curves present the predictive values of ceruloplasmin and serpin A7 measurements using targeted proteomic on maternal urine samples collected at 20–24 weeks of gestation (**A**) and at 30–34 weeks of gestation (**B**). The area under the ROC curves (AUC) were significant for ceruloplasmin at 20–24 weeks (AUC: 0.78; 95%CI: 0.58–0.97, *p* = 0.016); serpin A7 at 20–24 weeks (AUC: 0.75; 95%CI: 0.57–0.92, *p* = 0.028); ceruloplasmin at 30–34 weeks (AUC: 0.92; 95%CI: 0.82–1.00, *p* < 0.001); serpin A7 at 30–34 weeks (AUC: 0.77; 95%CI: 0.56–0.98, *p* = 0.024).

**Table 1 jcm-10-04679-t001:** Characteristics of our study populations.

	Discovery Study		Validation Study		Longitudinal Study	
	Cases (*n* = 12)	Controls (*n* = 12)	Cases (*n* = 12)	Controls (*n* = 12)	Cases (*n* = 10)	Controls (*n* = 20)
Maternal age (years)	29 (27–31)	29 (27–33)	29 (27–31)	30 (28–33)	31 (27–35)	30 (29–33)
BMI (kg/m^2^)	33 (28–36)	28 (26–33)	30 (28–35)	30 (25–34)	31 (29–33)	24 (22–26)
Caucasian	12 (100%)	12 (100%)	11 (92%)	12 (100%)	9 (90%)	20 (100%)
Gestational age at birth	34 (29–36)	39 (38–40)	34 (30–35)	40 (40–41)	35 (33–36)	40 (39–41)
Preeclampsia	12 (100%)	0 (0%)	12 (100%)	0 (0%)	9 (90%)	0 (0%)
Fetal growth restriction	6 (50%)	1 (8%)	2 (17%)	2 (17%)	4 (40%)	0 (0%)

Median (Interquartile range) or Number (percentage).

**Table 2 jcm-10-04679-t002:** Validation study by targeted proteomics of 21 urinary proteins in women with and without preeclampsia.

Protein Accession	Gene Name	Protein Description	Ratio	*p*-Value	Significance
P04217	A1BG	Alpha-1B-glycoprotein	4.85	0.00000	***
P43652	AFM	Afamin	6.89	0.00001	***
P25311	AZGP1	Zinc-alpha-2-glycoprotein	14.39	0.00005	***
P01024	C3	Complement C3	1.92	0.01029	*
P00915	CA1	Carbonic anydrase 1	6.20	0.03347	*
P00450	CP	Ceruloplasmin	6.55	0.00005	***
P02774	GC	Vitamin D-binding protein	11.31	0.00001	***
P19823	ITIH2	Inter-alpha-trypsin inhibitor heavy chain 2	6.35	0.01660	*
P02763	ORM1	Alpha-1-acid-glycoprotein 1	2.96	0.06649	
P19652	ORM2	Alpha-1-acid-glycoprotein 2	1.66	0.28634	
P01009	SERPINA1	Alpha-1-antitrypsin (serpin A1)	9.50	0.01026	*
P01011	SERPINA3	Alpha-1-antichymotrypsin (serpin A3)	3.83	0.00011	**
P08185	SERPINA6	Corticosteroid-binding globulin (serpin A6)	2.27	0.00039	**
P05543	SERPINA7	Thyroxine-binding globulin (serpin A7)	16.42	0.00013	**
P01008	SERPINC1	Antithrombin-III (serpin C1)	4.38	0.00000	***
I3L145	SHBG	Sex-hormone-binding globulin	4.35	0.00000	***
P02766	TTR	Transthyretin	4.32	0.00000	***
P02768	ALB	Albumin	2.57	0.00001	***
P69905	HBA1	Hemoglobin subunit alpha 1	3.74	0.02510	*
P68871	HBB	Hemoglobin subunit beta	24.62	0.00004	***
P02787	TF	Serotransferrin	2.82	0.02372	*

For each protein (Uniprot accession number given in the first column), average intensity ratio of PE over control groups (PE/CTL) is shown associated to its Student’s test statistical *p*-value and the corresponding significance. * *p* < 0.05; ** *p* < 0.001; *** *p* < 0.0001.

**Table 3 jcm-10-04679-t003:** Longitudinal study by targeted proteomics of 21 urinary proteins in women with and without preeclampsia.

	20–24 Weeks			30–34 Weeks		
Gene Name	Ratio PE/CTL	*p*-Value	Significance	Ratio PE/CTL	*p*-Value	Significance
SERPINA7	1.42	0.04247	*	1.62	0.00966	**
CP	1.81	0.04515	*	4.87	0.01784	*
AFM	1.29	0.33745		2.48	0.01821	*
ITIH2	1.14	0.66046		3.67	0.02808	*
TF	1.86	0.27263		8.42	0.03446	*
A1BG	1.19	0.50528		2.22	0.04651	*
SERPINA3	1.39	0.22978		1.95	0.05920	
GC	1.20	0.28638		1.97	0.06296	
ALB	1.49	0.20602		5.54	0.06954	
SERPINA1	1.36	0.24731		4.11	0.07165	
C3	1.07	0.85042		1.63	0.07776	
ORM1	1.69	0.42028		2.52	0.07781	
SERPINA6	1.05	0.84649		1.97	0.08995	
ORM2	1.35	0.49657		1.52	0.19926	
TTR	1.23	0.29396		1.48	0.21370	
HBA1	5.27	0.01524	*	55.40	0.30338	
HBB	5.43	0.02956	*	30.91	0.33029	
SERPINC1	0.99	0.97216		1.28	0.34586	
AZGP1	1.44	0.59655		1.19	0.61454	
CA1	0.62	0.55159		0.70	0.63019	
SHBG	0.92	0.58530		1.07	0.72452	

For each protein (Uniprot accession number given in the first column) at two-time points (20–24 or 30–34 weeks of gestation), average intensity of PE over control groups (PE/CTL) is shown associated to its Student’s test statistical *p*-value and the corresponding significance. * *p* < 0.05; ** *p* < 0.01.

## Data Availability

The data presented in this study are available on request from the corresponding author.

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
