# Peer review of "Proteomic Analysis of Maternal Urine for the Early Detection of Preeclampsia and Fetal Growth Restriction"

_jcm, 2021, doi:10.3390/jcm10204679_

Round 1

Reviewer 1 Report

This is an interesting study that identified some biomarker proteins for early detection/prediction of preeclampsia and fetal growth restriction. The study was well designed; the experiments were done properly; and the manuscript is generally well written. However, the paper needs some corrections.

  1. Lines 24, 29, 244, 249,,, should be “ceruloplasmin and serpin A”.
  2. Lines 50, 53, 113, 138,,,; should be “LC-MS/MS” as found in the Abstract.
  3. Line 54; 1,800 rather than 1800.
  4. Line 58; should be “serpin A1 and albumin”.
  5. Line 77; “Urine samples were collected on the day of the,,,,,,”?
  6. Lines96, 97, 99,,,,,; Add spaces between 50 and mM, 85 and ul, 0.8 and mM, 200 and ng; 1 and ug,,,,,,,,.
  7. Line 118; “the mass spectrometer was operated”?
  8. Line 132; “if it mets the”.
  9. Line 139; should be “biotinidase (BTD)”.
  10. Table 1; Caucasian in the case of validation “11(92%)”
  11. Figure 2 legend; “Heatmap of the expression of the 1,108 proteins extracted from the proteomic analysis”?
  12. Line 183; should be “SHBG; TF; see Table 1 for protein descriptions”.
  13. Line 189: “fold change” but not “Fold Change”.
  14. Line 200, 207,,,,,, p should be italic.
  15. Lines 221-222; they were abbreviated in lines 181-182.
  16. Table3; do not repeat Protein Accession and Protein Description (They are in Table 2), and make enough spaces between ratios and p-values.
  17. Line 262; How “many”?
  18. Line 267; delete “clearly” because PCA works so.
  19. Line 281; the authors should briefly explain what is ceruloplasmin and serpina 7 and how they work.

Reviewer 2 Report

In the study, the authors aim to assess the urine proteomic profiles in preecamplasia  and to identify biomarkers that could help in the early prediction. the study conducted. it is well done and structured and certainly can be a starting point for the development of new markers of this condition although the number of subjects involved in the study is  low, it should be expanded in the future to have even more statistical significance. some comments and suggestions for the authors :

  1. from the heatmap, it would appear that the proteins with higher intensity (up-regulated) are relative to the controls than samples affected by PE. generally high expressions of proteins are found in pathological phenomena, how do the authors explain this?
  2. Figure 2 it would be necessary to show in the heatmap the names of the proteins identified by MS or at least add an additional figure on the expression of the 22 selected proteins in order to lead the reader in the correct interpretation of the graph
  3. although it is clear that PE is a condition that only develops in the gestational period, It is also clear that pregnancy leads to hormonal and non-hormonal changes in women that can be a source of variation at the molecular level and that these changes can translate into changes in protein expression, including physiological ones it would be interesting to include non-pregnant women in these studies and to evaluate the expression levels of these proteins. what do the authors think about it?
  4. could the authors explain if the twenty proteins identified are correlated with each other (e.g. protein interaction network which could explain the trend of their expression) or involved in important pathways (pathway analysis)?
